# Extremely Low-Frequency Electromagnetic Stimulation (ELF-EMS) Improves Neurological Outcome and Reduces Microglial Reactivity in a Rodent Model of Global Transient Stroke

**DOI:** 10.3390/ijms241311117

**Published:** 2023-07-05

**Authors:** Amanda Moya-Gómez, Lena Pérez Font, Andreea Burlacu, Yeranddy A. Alpizar, Miriam Marañón Cardonne, Bert Brône, Annelies Bronckaers

**Affiliations:** 1BIOMED, UHasselt, Agoralaan, 3590 Diepenbeek, Belgium; amanda.moyagomez@uhasselt.be (A.M.-G.); bert.brone@uhasselt.be (B.B.); 2Biomedical Engineering Department, Facultad de Ingeniería Informática, Telecomunicaciones y Biomédica, Universidad de Oriente, Santiago de Cuba 90 400, Cuba; 3Centro Nacional de Electromagnetismo Aplicado, Universidad de Oriente, Santiago de Cuba 90 400, Cuba

**Keywords:** extremely low frequency electromagnetic stimulation, ischemic stroke, ischemia/reperfusion injury, microglia migration

## Abstract

Extremely low-frequency electromagnetic stimulation (ELF-EMS) was demonstrated to be significantly beneficial in rodent models of permanent stroke. The mechanism involved enhanced cerebrovascular perfusion and endothelial cell nitric oxide production. However, the possible effect on the neuroinflammatory response and its efficacy in reperfusion stroke models remains unclear. To evaluate ELF-EMS effectiveness and possible immunomodulatory response, we studied neurological outcome, behavior, neuronal survival, and glial reactivity in a rodent model of global transient stroke treated with 13.5 mT/60 Hz. Next, we studied microglial cells migration and, in organotypic hippocampal brain slices, we assessed neuronal survival and microglia reactivity. ELF-EMS improved the neurological score and behavior in the ischemia-reperfusion model. It also improved neuronal survival and decreased glia reactivity in the hippocampus, with microglia showing the first signs of treatment effect. In vitro ELF-EMS decreased (Lipopolysaccharide) LPS and ATP-induced microglia migration in both scratch and transwell assay. Additionally, in hippocampal brain slices, reduced microglial reactivity, improved neuronal survival, and modulation of inflammation-related markers was observed. Our study is the first to show that an EMF treatment has a direct impact on microglial migration. Furthermore, ELF-EMS has beneficial effects in an ischemia/reperfusion model, which indicates that this treatment has clinical potential as a new treatment against ischemic stroke.

## 1. Introduction

Stroke remains one of the main causes of death and acquired disability worldwide [1]. Ischemic stroke is the sudden obstruction of a blood vessel and represents more than 80% of total stroke cases. Current treatment strategies have as main objective to restore the blood flow using either recombinant tissue plasminogen activator (rTPA) [2] or mechanical thrombectomy, which involves an innovative surgical method to remove the blood clot. In either case, effective recanalization of the occluded blood vessel can result in harmful effects in the brain tissue, designated as ischemia reperfusion (I/R) injury. The mechanisms associated include mitochondrial dysregulation, increased oxidative stress, neuronal apoptosis [2,3,4,5], and an acute neuroinflammatory response, which is one of the processes determining the secondary progression of ischemic lesions [2,3,4,5] Herein, microglia, immune cells residing in the central nervous system, play a key role. Microglia migration and secretion of inflammatory markers are directly linked to increased brain injury in the acute phase after stroke [6]. Diverse therapeutic strategies for I/R injury have been explored in animal studies; however, the poor therapeutic success rates in clinical studies point out the critical need for new treatments for stroke and I/R injury [7].

Electromagnetic fields (EMFs) are magnetic fields produced by moving electrically charged particles and can be viewed as a combination of electrical and magnetic fields. EMFs, in their pulsed and sinusoidal forms, have been studied as a stroke treatment for several years [8]. They have been demonstrated to affect diverse biological processes such as cellular membrane interaction, intracellular calcium concentration, free radicals, and nitric oxide (NO) production [9,10,11,12], which explains the rationale for their application as a stroke treatment. Nevertheless, the cellular and molecular mechanisms triggered by sinusoidal EMFs have not been fully elucidated [8]. Our research group previously showed that in permanent stroke models, sinusoidal extremely low-frequency electromagnetic stimulation (ELF-EMS) reduced lesion size, improved neurological outcome, and enhanced cerebrovascular perfusion. It also increased eNOS activation in endothelial cells [13,14]. In reperfusion models, studies show a recovery of sensory motor functions [15,16], reduction in oxidative stress [17], and an effect on neuronal death and glia activation [18]. However, these results were observed after long periods of EMF exposure, which might be an obstacle to translate for clinical settings. In addition, the molecular and cellular pathways were not investigated.

In this research, we explored the cellular and molecular mechanism underlying the effect of a sinusoidal ELF-EMS (13.5 mT/60 Hz) in a treatment scheme applicable to the clinic (4 sessions of 20 min each) in an ischemia/reperfusion stroke model. First, we investigated the therapeutic effect of ELF-EMS in a gerbil model of ischemia/reperfusion (Figure 1). Secondly, the effect of ELF-EMS on microglia and neurons in vitro as well as in hippocampal brain slices was assessed. Finally, we focused on how ELF-EMS affect microglial migration and the production of inflammatory markers.

## 2. Results

### 2.1. ELF-EMS Improves Outcome after Transient Global Cerebral Ischemia in Gerbils

Cerebral ischemia/reperfusion was achieved by transient occlusion of both common carotid arteries. Animals were then either sham-exposed or treated with ELF-EMS (13.5 mT/60 Hz/20 min) for 4 consecutive days as previously established by our research group [13,14]. ELF-EMS significantly improved the neurological score of treated ischemic gerbils 3 days (8.0 ± 0.8 vs. 2.9 ± 0.9) and 7 days (7.67 ± 1.0 vs. 2.13 ± 1.0) after surgery (Figure 1C).

In the open field test (Figure 1D), ELF-EMS-treated gerbils showed a better behavioral outcome than the control animals, as their rearing frequency was significantly higher (73.28 ± 18.90 vs. 128.8 ± 19.54). The ELF-EMS-treated group had a rearing frequency comparable to non-ischemic mice (210.40 ± 32.65) (dotted line). On the other hand, the ELF-EMS-treated group presented a less anxious behavior than the sham-treated group, with a preference to spend more time in the open areas of the elevated plus maze (95.3 ± 22.5 s vs. 185.6 ± 17.9 s). In addition, the tendency of the ELF-EMS-treated group to explore the open areas was similar to non-ischemic mice (Figure 1E, dotted line).

### 2.2. ELF-EMS Affects Neurons and Glia in Transient Global Cerebral Ischemia

To elucidate if the improvement in neurological outcome by ELF-EMS was associated with an improved neuronal survival, immunofluorescence staining using NeuN was performed to quantify neurons in the CA1 and CA2 regions of the stratum pyramidale. The number of NeuN+ cells were significantly lower in the sham-exposed group compared to the ELF-EMS-treated group 7 days after the surgery (49.43 ± 13.14 vs. 102.32 ± 19.62). At earlier time points, no differences in the number of NeuN+ cells were found between the groups (Figure 2A,B). In contrast, GFAP+ cells (astrocytes) were significantly higher in the sham-exposed group at day 7 (23.10 ± 2.52 vs. 16.44 ± 1.74), with no difference observed in the previous time points (Figure 2C,D), meaning that ELF-EMS reduces astrocyte activation.

In addition, we assessed the number of Iba1+ cells, which is a marker for microglia and macrophages. The presence of Iba1+ cells in the hippocampus gradually increased in time (Figure 2D,E), and it was significantly lower after ELF-EMS treatment since day 3 (16.20 ± 1.53 vs. 12.11 ± 1.00). At day 7, the observed amount of Iba1+ cells in the ELF-EMS-treated group was still significantly lower (27.32 ± 3.81 vs. 15.69 ± 2.22), suggesting a possible inhibitory effect of ELF-EMS on microglia/macrophage migration or proliferation.

### 2.3. ELF-EMS Does Not Affect Neuronal Survival In Vitro but in OHSC

Next, we analyzed whether the ELF-EMS had a direct effect on neurons. Therefore, cell survival and caspase 3/6 production were quantified in SHSY5Y cells and primary neurons were submitted to 4 h OGD and treated with 13.5 mT/60 Hz for 20 min. After 24 h, confluence and apoptosis in OGD-cell cultures was not statistically different between untreated and ELF-EMS-treated SHSY5Y or primary neurons (see Appendix A). Next, we used hippocampal brains slices submitted to 1 h OGD and 7 days of reperfusion. Four ELF-EMS treatments were applied, and neuronal survival was quantified using PI staining 1, 3, and 7 days after OGD. PI+ area was significantly lower in the ELF-EMS-treated group compared to the untreated OGD group (Figure 3A,B). Contrary to what was observed in vivo, this difference started to be evident 3 days after OGD (5.09 ± 0.32 vs. 4.17 ± 0.23) and the difference increased after 7 days (OGD; 6.98 ± 0.51 vs. 5.30 ± 0.38).

### 2.4. ELF-EMS Does Not Affect Microglia Proliferation

Post-ischemic microglia proliferate and migrate to the lesion site [19]. In order to elucidate which process is affected by ELF-EMS, we used Ki67 as the proliferation marker in 1-day samples of gerbil brains. There was no difference in mean relative fluorescence between the stroke and ELF-EMS-treated group, (Figure 4A,B), leading to the conclusion that ELF-EMS does not affect proliferation of brain cells. To confirm that ELF-EMS does not affect proliferation of microglia, we used OGD/R as the stroke model in vitro with BV2 cells, a microglia cell line. Cells were incubated in OGD conditions (5% CO_2_ and 1% O_2_) for three hours and proliferation during reperfusion was monitored for 48 h. No difference in confluence was observed between groups during the 48 h of the experiment (Figure 4C). These results suggest that ELF-EMS does not affect proliferation in these conditions.

### 2.5. ELF-EMS Decreases Microglia Migration

#### 2.5.1. LPS- and ATP-Induced Migration in BV2 Cells

To further analyze if ELF-EMS affected microglia migration, BV2 cells were submitted to different activating stimuli and migration was assessed by both the scratch and transwell assay. First, LPS (100 ng/mL) was used as an inflammatory stimulus to increase microglia chemokinesis [20]. LPS treatment induced BV2 microglial cell migration in the scratch assay. In addition, ELF-EMS significantly decreased this type of migration compared with LPS treatment alone (1.56 ± 0.11 vs. 0.97 ± 0.22) (Figure 5A,B). Next, we investigated whether ELF-EMS regulated BV2 microglial chemotaxis in a transwell assay. LPS (100 ng/mL) was added as a chemoattractant. ELF-EMS also significantly decreased LPS-oriented transmigration (1.50 ± 0.11 vs. 1.22 ± 0.08) (Figure 5E,G). Even though LPS is a widely used neuroinflammatory stimulus in vitro [20,21,22], it is usually associated with non-sterile inflammation, on the contrary to what happens in the brain after an ischemic insult. Therefore, we also applied ATP (300 µM) to induce migration, as this is released by dying neurons during stroke. Similar to the LPS experiments, ATP-induced migration in the scratch assay was significantly reduced by the ELF-EMS treatment (1.42 ± 0.06 vs. 0.84 ± 0.05) (Figure 5C,D). Moreover, transmigration towards ATP was also reduced in the ELF-EMS cells compared to the sham-treated ones (1.78 ± 0.12 vs. 1.19 ± 0.12) (Figure 5F,H).

#### 2.5.2. ELF-EMS Decreases ATP Induced Migration in Primary Microglia

To confirm the effect of ELF-EMS on migration, its effect on transwell migration of primary mouse microglia evoked by ATP was investigated. Again, ELF-EMS had an inhibitory effect on ATP-induced transmigration of mouse primary microglia (2.47 ± 0.31 vs. 1.37 ± 0.15) (Figure 5I,J).

### 2.6. ELF-EMS Decreases Microglia Reactivity in Organotypic Hippocampal Slices Culture

Reactivity to OGD of microglia in CX3CR1WT/eGFP mice hippocampal brain slices after OGD/R was assessed 1, 3, and 7 days after stroke stimulus. OGD boosted microglia reactivity to the hippocampus compared to the control, where the area occupied by microglia ranged from 14.49% (day 1) to 33.17% (day 7). Slices that were subjected to OGD exhibited an increasing amount of microglia reactivity in the hippocampus, detectable during a 6-h period (Figure 6A) in the three different time points. Moreover, treatment with ELF-EMS reduced microglia reactivity compared to the OGD alone group, and this was statistically significant in the last 6-h time point at day 1 (64.18 ± 2.96% vs. 50.72 ± 3.53%), day 3 (75.40 ± 0.84 vs. 57.14 ± 06.33), and day 7 (93.37 ± 1.35 vs. 67.48 ± 8.58) (Figure 6B,C). These results confirm that ELF-EMS reduces microglia reactivity under OGD circumstances.

### 2.7. ELF-EMS Affects Inflammation Markers in OHSC but Not in Cell Cultures

Because microglia are one of the main actors in the CNS immune response, we investigated if ELF-EMS did not only affect migration but also the expression of inflammation markers. Nitrite concentration, an indirect measure of NO, was significantly increased as a result of adding LPS and IFN-γ to BV2 cell culture. However, treatment with ELF-EMS did not affect nitrite concentration either in basal conditions or after the ischemic stimulus (Figure 7A). Similarly, ELF-EMS did not regulate the increased gene expression of inflammatory cytokines *Tnf-α*, *IL-1β* and the enzyme *iNos*, nor did it alter the expression of *Arg-1* (Figure 7B). In contrast, ELF-EMS prevented an increase in nitrite in OHSC submitted to OGD (Figure 7C). Moreover, qRT-PCR analysis showed that ELF-EMS significantly reduced the increase in OGD-upregulated anti-inflammatory (*Cd163*) and pro inflammatory markers (*iNos*, *Il-1β*). (Figure 7D,E). Expression of *Tnf-α* and *Arg-1* was not statistically significant different among groups.

## 3. Discussion

This study shows that a clinical treatment scheme with sinusoidal ELF-EMS improves neurological outcome in an ischemia/reperfusion stroke model, and this is linked with neuronal survival and reduced glia reactivity. Moreover, it decreases microglia migration in vitro and microglia activation ex vivo.

Throughout cerebral ischemia-reperfusion, a series of pathological cascades are activated that directly or indirectly cause apoptosis/death of neurons and lead to neurological deficits [23]. In this study, we show that ELF-EMS rescued neurological deterioration resulting from ischemia/reperfusion injury, mainly it improved neurological function. I/R injury causes a decrease in rearing frequency in ischemic animals that is recovered by ELF-EMS to levels similar to non-stroke animals.

Moreover, the time in the open spaces of the elevated-plus maze was also lower in animals treated with ELF-EMS, which can be related to a decrease in anxiety behavior, and which has been linked to alterations in the hippocampus [24,25]. This structure is sensitive to ischemic hypoxia as it is affected by cell necrosis, structural damage, and cell reduction after I/R injury [26]. Additionally, transient global cerebral causes neurodegeneration in the hippocampal CA1 and CA2 region. This loss in neuronal density starts to be evident at day 3 or 4 after stroke [27,28] and can still be observed >30 days after reperfusion [29,30]. We found that I/R-induced neuronal loss in this area was rescued by ELF-EMS after 7 days of reperfusion, which is in line with previous reports using other types of EMF in different stroke models [17].

Glial activation (i.e., increase in GFAP+ cells) is another result of I/R injury [31,32] and accelerates ischemic neuronal damage and triggers inflammatory responses [33]. Therefore, reduced astrocyte activation is connected to neuroprotection after brain ischemia [29,31,32]. Accordingly, we found that ELF-EMS decreases astrocyte activation within 7 seven days after injury, while others reported an increase of GFAP+ cells after EMF treatment [18]. The latter study applied a longer exposure period and different magnetic field induction and frequency, indicating that a clear characterization of the different magnetic field parameters is necessary.

During transient ischemia, microglial activity is first reduced due to hypoxia presenting thickened and reduced processes and decreased extension and retraction rates. After reperfusion, microglia become reactive. Diverse research groups have found an increase in microglia activation in the hippocampus, particularly the CA1 region, after ischemia [27,34]. This increased reactivity starts as early as 24 h after ischemic injury [27,28,29] and it is closely associated with the development of neuronal cell death.

An early and transient microglial reaction occurs throughout most of the hippocampus within 24 h of ischemia [27]. This overactivation of microglia conversely amplifies inflammation, leading to neuronal death [35,36]. Our data show an increase in Iba1+ cells in the hippocampus of gerbils submitted to t-BCCAO. Treatment with ELF-EMS decreased this amount, which is associated with a protective effect as reported by other authors with other EMF approaches [29,33]. However, similar to astrocytes, Raus et al. found that continued exposure to EMF for over 7 days increased microglia activation in models of transient global ischemic stroke [18]. This highlights the differential effect of EMF depending on its settings (such as frequency, duration, time of application after stroke, magnetic flux intensity). Moreover, we confirmed the decreasing effect of four sessions of 20 min of ELF-EMS on microglia reactivity in OHSC.

Microglial transformation after stroke might include proliferation, polarization, and increase migration. Here we showed that the ELF-EMS-induced decrease in Iba1+ cells in the hippocampus is not caused by a reduction in proliferation. The massive expansion of microglia in the hippocampal CA1 region can be observed after transient global ischemia, but this type of ischemia does not induce significant proliferation of microglia in cerebral cortices [37], as we confirmed by Ki67 staining. Moreover, in another model of hippocampal lesion, where injury was induced in OHSC with NMDA treatment, no difference in proliferation was noted. However, an increase in microglial migration was observed using time-lapse imaging, resulting in an increase in microglia in the hippocampus [38], which is in concurrence with our data showing that ELF-EMS reduces microglial migration in monocultures.

In OHSC, 1 h OGD promoted microglia reactivity, which was inhibited by ELF-EMS as early as 24 h after OGD. Additionally, PI staining in OHSC showed that OGD induced neuronal death without affecting microglia or astrocytes as previously reported [39,40]. In this assay, unlike in neuronal cell cultures, ELF-EMS promoted neuronal survival from day 3, i.e., after the time point at which a decrease in microglial reactivity was observed. Firstly, this indicates that microglial reactivity is not triggered after neuronal loss, but microglia sense neuronal damage early before cell death. It has been shown that microglia react to reduced or increased neuronal firing [41] and stress [42]. Secondly, this indicates that microglia activation further increases neuronal damage. Their hyperactivation after stroke reduces their clearance capacity, which eventually results in neuronal death and breakdown of the BBB [6].

Our results also show that the effect of ELF-EMS on neuroprotection is indirect by affecting other cells of the neurovascular unit. Previous results from our group with the same ELF-EMS settings (but in a permanent model of ischemia) showed the effect of ELF-EMS on circulation and its importance to lesion size improvement [13]. Interestingly, neurons are also recovered in OHSC, where there is no circulation, pointing out the important role of glial cells in neuronal survival. However, brain endothelial cells also exert a neuroprotective role besides vasodilation and angiogenesis. They secrete neurotrophins such as brain-derived neurotrophic factor (BDNF), which are involved in ensuring synaptic plasticity and neuronal survival [43] and have been linked to recovery after stroke in animal models [44,45].

Our work significantly demonstrates that ELF-EMS decreases microglial migration. As far as we know, this is the first time this effect of sinusoidal EMF has been reported. Neuronal injury induces a reactive state in microglia, which in turn show a directional migration toward the site of injury using chemoattractant gradient as a directional cue [46]. Decreased migration has been related to decreased microglial activation, inflammation, and improvement in neurological outcome [20,47].

Among the receptors involved in microglial migration are ionotropic (P2X) and metabotropic (P2Y) purinergic receptors. The inhibition of P2Y12, which is involved in migration and inflammation [48], increases neuron viability in the CA1 region of the hippocampus and decreases immune responses following BCCAO [49]. Blocking P2X7, which mediates the proliferation and migration of microglia [50], reduces stroke lesion, and inhibition of P2X4 [51] has shown a neuroprotective effect after ischemic stroke [52]. Moreover, it has been found that magnetic and electric fields affect the activation of some purinergic receptors in mesenchymal and tumor cells [53,54]; therefore, further analysis of the implication of these receptors on microglia migration and the effect of EMF are highly required.

The mechanosensitive ion channel Piezo1 is also involved in microglia migration. The activation of this calcium channel has been shown to decrease microglia migration in vitro and in vivo [55]; however, its activation on endothelial cells produces an increase in migration [56]. Interestingly, stimulation of endothelial cells with EMF promotes proliferation and migration [57,58], suggesting a differentiated effect of EMF could be linked to the activation of this channel. Piezo1 is also required for the regulation of endothelial NO formation [59], which is produced by the enzyme eNOS [60], which is activated by ELF-EMS [13].

Another ion channel that might be involved in ELF-EMS-induced microglial activations is TRPV4. Inhibition of this ion channel induces changes in the microglia cytoskeleton, decreasing their motility [61]. Interestingly, in a model of traumatic brain injury, inhibition of TRPV4 protected neurons by reducing the expression levels of cytokines IL-1β and TNF-α [62]. In this model, similar to I/R, injury was more noticeable in the CA1 region, where astrocytic and microglial activation was observed prior to neuronal apoptosis [63]. Radio frequency EMF have shown to induce changes in the gating properties of TRPV4 [64,65], but no effects in the extremely low-frequency range have been reported so far. Similar to Piezo 1, the role of TRPV4 in microglia migration and activation in pathological conditions has to be further studied, as well as the possible effect of ELF-EMF under the clinical treatments scheme.

## 4. Materials and Methods

### 4.1. Sinusoidal ELF-EMS

ELF-EMS was generated by using a Magnetic Stimulator NaK-02 with a coil (ferromagnetic core radius 16 mm; wire diameter 0.20 mm; 950 turns) manufactured by our collaborators from Cuba (F.G. González, M.C. Cardonne and L.P. Font of Centro Nacional de Electromagnetismo Aplicado (CNEA)). The generated magnetic field was a continuous sinusoidal current with a frequency of 60Hz (in the extremely low-frequency range) and a magnetic intensity of 13.5 mT without intervals, hence a non-pulsed electromagnetic field [13,14]. The produced magnetic field and possible magnetic interference from the environment was measured using a calibrated PCE-MFM 3000 gaussmeter (PCE Instruments, Enschede, The Netherlands).

### 4.2. Animals

Mongolian gerbils (6-month-old males) from the National Center for Production of Laboratory Animals (CENPALB, La Havana, Cuba) were used for the behavioral and immunofluorescence analysis. Gerbils were housed under controlled conditions (humidity 55–60%; temperature 20 ± 2 °C and 12-h light-dark cycle) with food and water ad libitum. For primary microglia isolation and organotypic hippocampal slice culture, C57BL/6 mice wild-type (WT) and CX3CR1WT/eGFP were used with the same controlled conditions. The experiments were performed according to the guidelines defined in the “Principles of laboratory animal care” (NIH publication No. 86-23, revised 1985), the EU Directive 2010/63/EU as well as the specific Belgian law (Belgian law of animal welfare and Royal Decree of 29 May 2013). All experiments were approved by the Ethical Committee for Animal Experimentation (ECAE) of Hasselt University or by the Experimental Animal Ethic Committee of CENPALAB.

### 4.3. Ischemia/Reperfusion Model

The transient bilateral common carotid artery occlusion (t-BCCAO) was performed in Mongolian gerbils as described previously [16]. Gerbils present an incomplete circle of Willis; therefore, occlusion of both common carotid arteries results in global cerebral ischemia [66,67]. Briefly, both common carotid arteries were exposed and clamped for 15 min (Figure 1A). Next, animals were randomly assigned to either sham or ELF-EMS treatment using blocking randomization. From this point on, ELF-EMS-stimulated animals will be referred to as the treated group and the sham-treated ones as stroke or ischemic. The treated group were repeatedly exposed to ELF-EMS (13.5 mT/60 Hz) for 20 min on 4 subsequent days, with the first exposure 1 h after injury (Figure 1A,B).

### 4.4. Neurological and Behavior Examination

Neurological deficits in gerbils were examined regularly 1, 3, and 7 days after t-BCCAO. The deficits were scored as described by Menzies et al. [68] as previously stated [14]. Clinical signs of stroke such as piloerection, hypotony, and problems in marching and posture were quantified, meaning that animals with greater deficits had higher scores. Only animals with scores higher than 3 (medium to severe signs) at 24 h post-stroke were included in the analysis. In the ischemic and treated group, three and two animals were left out, respectively. Locomotor activity and anxiety level were assessed at day 7 after t-BCCAO with the open field test and the elevated-plus maze, respectively. The number of rears up in the open field and the amount of time in open spaces in the plus-maze were recorded and quantified in a 5-min time lapse (Figure 1B). Non-ischemic and untreated animals were also included. In addition, at 24 h, 3 days, and seven days, animals (n = 5–8) were sacrificed to perform immunohistochemistry on brain samples.

### 4.5. Immunofluorescence Staining of Brain Samples

At 24 h, 3 days, and seven days, animals (n = 5–8) were anesthetized and transcardially perfused, first by using phosphate buffer saline (PBS) and then 4% paraformaldehyde (PFA). The brain was isolated, fixed with 4% PFA in PBS for a maximum of 24h, and embedded in paraffin. For immunohistochemistry, 7 µm slices were cut and deparaffinized using xylene and lowering grades of ethanol, rinsed with PBS, permeabilized with 0.05% Triton X-100 (93426 Fluka, Charlotte, NC, USA) blocked with 10% protein block serum-free (X0909, Dako, Glostrup, Denmark), and incubated with primary antibodies overnight at 4 °C. The primary antibodies, mouse anti-NeuN antibody (1:400, MAB377, Millipore, Billerica, MA, USA) for neurons, rabbit anti-Iba1antibody (1:250, 019-19741, Wako Pure Chemical, Tokyo, Japan) for microglia/macrophages and mouse anti-glial fibrillary acidic protein (GFAP) antibody (1:300, G3893, Sigma-Aldrich, St. Louis, MO, USA) for astrocytes were used. The slices were rinsed with PBS and incubated with secondary antibodies for 1 h. As secondary antibodies, Alexa Fluor 488-labeled goat anti-rabbit IgG antibody, Alexa Fluor 488-labeled donkey anti-mouse IgG antibody, and Alexa Fluor 568-labeled donkey anti-mouse IgG antibody (1:500 each; Invitrogen, Waltham, MA, USA) were used. After rinsing in PBS, cultures were stained with DAPI nucleus dye. Three images of the hippocampus region of six samples per animal were obtained with an inverted fluorescence microscope (Leica DM4000 B LED, Wetzlar, Germany) equipped with a CCD camera (Leica, DFC 450 C, Wetzlar, Germany) at 20×.

### 4.6. Cell Culture

BV2 cells, an immortalized murine microglial cell line, and SHSY5Y cells, an immortalized human neuroblastoma cell line were both cultured in DMEM (41966-029, Gibco, Paisley, Scotland) supplemented with 10% fetal bovine serum (Gibco, 26140-079), 100 U/mL penicillin, and 100 μg/mL streptomycin (1% P/S; Sigma-Aldrich, P4333, Merck, Darmstadt, Germany) in a 37 °C humidified atmosphere with 5% CO_2_. Cells were sub-cultured every 2 or 3 days when reaching 80% confluence and seeded at cell densities appropriate for each assay.

Primary microglia from P21 WT or CX3CR1WT/eGFP mouse pups were obtained by Magnetic activated cell sorting (MACS) as previously described in [69]. Briefly, pups’ brains were isolated, and the hemispheres were transferred to ice-cold DMEM containing 1% P/S. Brains were mechanically homogenized, enzymatically digested with Papain, and DNase and sieved through a cell strainer (70 μm). Mononuclear cells were isolated using a 30–70% Percoll gradient. MACS was performed according to the manufacturer’s guidelines (MACS, Miltenyi Biotec, Bergisch Gladbach, Germany) using MS columns (130-042-201) and CD11b microbeads (130-049-601). Microglia were seeded in poly-D-lysine (PDL) and collagen-coated plates at specific cell densities, depending on the assay. Cells were cultured in DMEM supplemented with 10% fetal bovine serum, 10% horse serum (Gibco, 26050-088) and 1% P/S (10:10:1 medium) in a 37 °C humidified atmosphere with 8.5% CO_2_ for 5 to 7 days before conducting any experiment.

Primary neurons were isolated from P1 WT mice pups. Brains were dissected and the hemispheres without the meninges were enzymatically digested with Trypsin and DNase and mechanically homogenized. Supernatant with the neurons was centrifuged and resuspended in Neurobasal medium (Gibco, 21103049) supplemented with 2% B27 (Gibco, A1486701), 0.5% Glutamine (Sigma, G751) and 0.1% P/S. Neurons were seeded in PDL-coated plates at specific cell densities depending on the assay. Cells were cultured in a 37 °C humidified atmosphere with % CO_2_ for 8 days before any experiment was conducted, and the medium was changed every 3 days.

### 4.7. Organotypic Hippocampal Slice Cultures

Organotypic hippocampal slice cultures (OHSCs) were prepared as described in [38]. Briefly, P5-P7 WT or CX3CR1WT/eGFP-mouse pups were sacrificed, and the brains were removed and placed in cold HBSS (Gibco). The hippocampi were extracted and cut into 300 µm slices with a tissue chopper (McIlwain, Dublin, Ireland). Slices were transferred onto 30-mm Millicell-CM insert membranes or 12-mm Millicell-CM insert membranes (pore size, 0.4 mm; Millipore, Bedford, MA, USA). The culture medium of “Organotypic hippocampal slice cultures” was composed of 50% MEM medium (12360-038, Gibco), 25% HBSS, and 25% heat-inactivated horse serum (16050-122, Gibco), supplemented with 6.5 mg/mL D-glucose (A24940-01, Gibco), 2 mM Glutamax (35050-038, Gibco), 1 M HEPES (15630-080, Gibco), and 1% P/S. The slices were maintained for 7–9 days before any experiment was conducted [70].

### 4.8. In Vitro ‘Oxygen Glucose Deprivation/Reperfusion’ (OGD/R) Experiments

Ischemia/reperfusion was simulated in vitro by subjecting cells and OHSCs to hypoxia and glucose-free media (OGD). First, cells and OHSCs were placed in glucose-free DMEM medium (11966-025, Gibco) for 15 min and placed in an O_2_/CO_2_ incubator (Sanyo, MCO-18M, Osaka, Japan) at 37 °C, 1% O_2_, and 5% CO_2_. Immediately after the OGD procedure, glucose-free medium was replaced with standard media and cultures were incubated under normoxic conditions to mimic reperfusion. BV2 and SYHY5Y cells were kept in OGD/R for 2/24 h. Primary neurons and OHSC were in OGD for 1 h and kept in reperfusion for 24 h and 7 days, respectively. In all cases, cells and OHSC were stimulated or sham-treated with ELF-EMS following the same in vivo scheme. Sham-treated cultures (from now on untreated or OGD) were taken out of the incubator for a similar amount of time to the cultures receiving ELF-EMS (treated), but without switching on the current in the coil.

Survival of BV2, SYSY5Y, and primary neurons and caspase activity was monitored with the Incucyte Live-Cell Analysis system (Sartorius, Goettingen, Germany). Phase contrasts images were taken to assess confluence (%), while apoptosis was quantified with Incucyte green dye Caspase 3/7 (1:1000). Confluence was normalized to the starting time point (t = 0 h) and the ratio Green Confluence/Phase Confluence was calculated.

### 4.9. Nitrite Measurement

To assess nitrite production, BV2 cells were seeded in a 24-well plate (200,000 cells/well). After 24 h, medium was replaced by 300 µL of medium containing Lipopolysaccharide (LPS, 100 ng/mL), with Interferon gamma IFN-γ, 20 ng/mL) as inflammation stimuli [22]. Half an hour after inflammation stimuli, cell culture was stimulated or not with ELF-EMS (13.5 mT/60 Hz for 20 min). After 24 h, medium was collected, centrifuged, and stored at −80 °C until nitrite measurement was performed. Similarly, supernatant of OHSCs was collected 7 days after I/R. Nitrite levels were measured using the Griess Regent System (Promega Benelux B.V, Leiden, The Netherlands) according to the manufacture’s guidelines.

### 4.10. RNA Extraction and PCR Analysis

BV2 cells were cultured in 12-well plates (200,000 cells/well). The culture and treatment protocol are described above (see Section 4.9). Six hours after treatment, cells supernatant was removed, cells were trypsinized and centrifuged, and pellet was conserved at −80 °C until RNA extraction. For OHSCs, 6 slices per sample were collected and stored in Quiazol at −80 °C. Before RNA isolation, tissue was homogenized, mixed with chloroform centrifuged for 15 min, and the RNA layer was collected. RNA extraction was performed using a RNeasy Mini Kit (74104, QIAGEN, Hilden, Germany) according to the manufacturer’s instructions. RNA (500 ng measured with NanoDrop Technology, Waltham, MA, USA) was reverse transcribed into cDNA using the SuperScriptTM III First Strand Synthesis SuperMix from QuantaBio (Beverly, MA, USA). qRT-PCR was performed using quantitative PCR systems (Applied Biosystems Thermo Fisher Scientific, Waltham, MA, USA) with corresponding primers (Table 1) and a fluorescent dye (RT2 SYBR Green FAST Mastermix, Applied Biosystems, Waltham, MA, USA). All primers were purchased from IDT or EUROGENTEC. The cycle threshold (CT) was normalized to Actin-B and GAPDH expression. The expression levels of mRNAs are reported normalized to control. ELF-EMS was generated by using a Magnetic Stimulator NaK-02 with a coil (ferromagnetic core radius 16mm; wire diameter 0.20 mm; 950 turns) manufactured by our collaborators from Cuba (F.G. González, M.C. Cardonne and L.P. Font of Centro Nacional de Electromagnetismo Aplicado (CNEA)). The generated magnetic field was a continuous sinusoidal current with a frequency of 60Hz (in the extremely low frequency range) and a magnetic intensity of 13.5 mT without intervals, hence a non-pulsed electromagnetic field [13,14]. The produced magnetic field and possible magnetic interference from the environment was measured using a calibrated PCE-MFM 3000 gaussmeter (PCE Instruments, Enschede, The Netherlands).

### 4.11. Migration Experiments

#### 4.11.1. Scratch Assay

Scratch assay was performed using silicone Ibidi inserts. Inserts were place in 12-well plate and 70 µL of standard BV2 medium with 35,000 cells were seeded in each compartment. After 24 h, inserts were removed, and the wells were filled with serum-free medium with or without LPS (100 ng/mL) [20] or ATP (300 µM) [71]. Cells were sham-treated or stimulated with ELF-EMS (20 min of 13.5 mT/60 Hz) 30 min after seeding. They were allowed to migrate for 24 h. Four pictures per sample were taken (20× magnification) 24 h after ELF-EMS with a Zeiss Model AxioVert 100 Inverted Microscope. The area occupied by migrated cells was quantified using Image J (version 1.53i) software [72]. Results are presented normalized to control culture.

#### 4.11.2. Transwell Migration Assay

Cell transmigration assays were performed using Corning Transwell membrane filters (8 µm pore size, Corning Costar, New York, NY, USA). Inserts on both sides were coated with fibronectin (10 µg/mL). BV2 cells were placed overnight on serum-free medium, and the next day seeded at a density of 104 cells/insert in 100 µL serum-free medium. Bottom wells were filled with serum-free medium (0%) (Control), with LPS (100 ng/mL), or ATP (300 µM) as chemoattractant. BV2 cells were allowed to migrate for 6 h. Primary microglia were seeded at a density of 10^4^ cells/insert and were allowed to migrate for 24 h. Bottom wells were filled with 10:10:1 complete medium with or without ATP (300 µM) [71]. Cells were sham-treated or stimulated with one session of ELF-EMS (20 min of 13.5 mT/60 Hz) 30 min after seeding. Once migration time was completed, cells were fixed for 5 min in 4% PFA, washed with PBS, and stained for 2 min with 0.05% crystal violet. Migrated cells were at the bottom side of the insert, thus the upper side of the inserts were swapped with cotton buds. From each insert, four pictures of the bottom side were taken on 10× magnification with a Zeiss Model AxioVert 100 Inverted Microscope. The area occupied by migrated cells was quantified using Image J software. The results are presented normalized to control culture (no ATP, no EMF treatment).

### 4.12. Measurement of PI and Immunostaining of OHSC

In OHSC, neuronal death was monitored 24 h, 3 days, and 7 days after OGD. Propidium iodide (PI) was added to the culture medium (5 µg/mL) 1h before live-imaging. PI fluorescent images were acquired with a confocal laser scanning microscope (LSM880; Zeiss, Oberkochen, Germany). Excitation at 543 nm and an emission band-pass filter of 585–615 nm with the EC Plan-Neofluar 10×/0.30 M27 objective were used to visualize PI-stained cells. Z-stack acquisition was used to optically section the OHSC at a thickness of 2 µm. The culture medium was replaced with fresh medium containing PI after each observation.

At day 7, slices were fixed with 4% PFA in PBS for 15 min and washed 3 times with PBS. OHSCs were pre-incubated in 5% normal horse serum with 0.3% Triton X100 in PBS for 3 h at room temperature and incubated with primary antibodies overnight at 4 °C. The primary antibodies used where mouse anti-NeuN antibody (1:400, MAB377, Millipore) for neurons, rabbit, anti-Iba1 antibody (1:250, 019-19741, Wako Pure Chemical) for microglia/macrophages, mouse anti-glial fibrillary acidic protein (GFAP) antibody (1:300, G3893, Sigma) for astrocytes, goat anti CD163 (1:50, 553142, BD Pharmingen, San Diego, CA, USA) for ant-inflammatory microglia and rat anti MHC II (1:50, sc-59322, Santa Cruz, Dallas, TX, USA) for pro-inflammatory microglia. Secondary antibodies were the same as the ones used for IHC. PI/Iba-1⁄ NeuN and PI⁄Iba-1⁄GFAP triple staining were analyzed with a confocal laser scanning microscope (LSM880; Zeiss, Germany) with EC Plan-Neofluar 20×/0.30 M27. Quantification CD163+ and MHC II+ cells were done with ImageJ software.

### 4.13. Microglia Reactivity in OHSC

To assess microglial reactivity in OHSC, CX3CR1WT/eGFP mice were used. In the CNS, microglial cells are the only cells expressing CXC3R1. Heterozygous animals are needed so that all microglia express one allele for the CX3CR1 receptor and eGFP under the second allele of CX3CR1 [73]. OHSC from CX3CR1WT/eGFP mice were cultured in 12-mm Millipore inserts for analyzing microglia migration. The slices were submitted to the OGD and the treatment protocol stated previously. At day 1, 3, and 7, slices were imaged. Ten pictures in a 6-h interval were taken and eGFP+ cells in three regions of the hippocampus were quantified using ImageJ.

### 4.14. Statistical Analysis

Data analysis was performed using GraphPad Prism 9 statistical software (GraphPad Software, La Jolla, CA, USA). All experiments were repeated at least three independent times, and n indicates the number of individual mice, independent in vitro experiments, or slices used in the study unless stated otherwise. For statistical analysis, in any experiment with only two groups, a one-tailed *t* test was used. For experiments with more than two groups, a one-way analysis of variance (ANOVA) with Tukey post hoc test was used. Data are represented as mean values  ±  SEM; *p*  <  0.05 was considered significant.

## 5. Conclusions

In conclusion, we showed that ELF-EMS treatment improves the neurological outcome in models of I/R. This treatment has an effect first on microglia activation and, subsequently, on neuronal survival and astrocyte activation. Furthermore, ELF-EMS decreases microglial migration in vitro and modulates the expression of inflammation markers. These findings further support the rationale for ELF-EMS as a treatment for stroke.

## Figures and Tables

**Figure 1 ijms-24-11117-f001:**
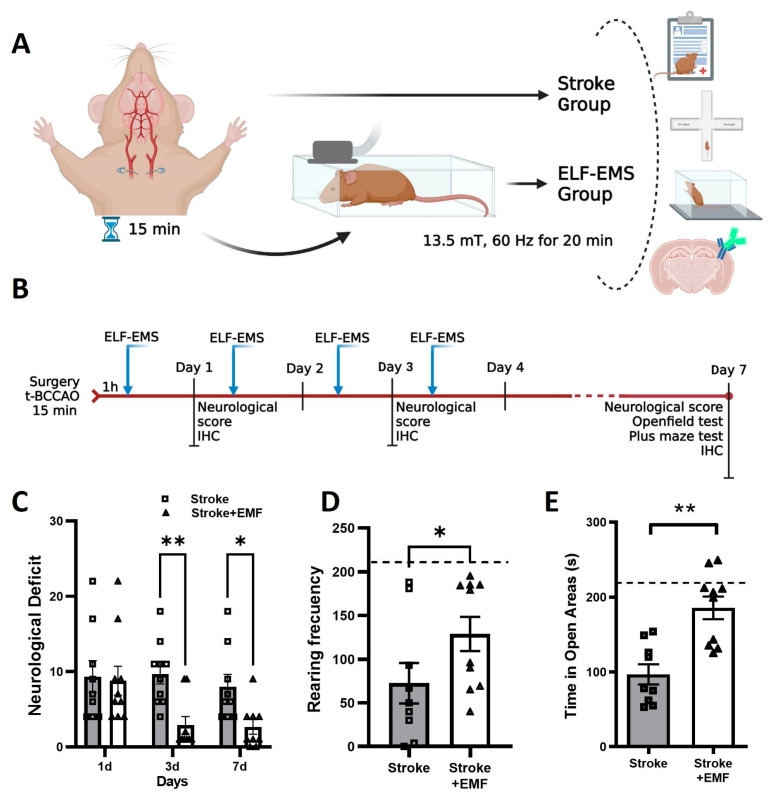
ELF-EMS improves the outcome of gerbils submitted to transient bilateral common carotid artery occlusion (t-BCCAO). (**A**) Schematic representation of the in vivo experiment. Both common carotid arteries were occluded for 15 min. After reperfusion animals, were randomized into two groups: Sham-treated (stroke) and ELF-EMS treated (Stroke + EMF) with 13.5 mT/60 Hz for 20 min and several behavior analysis and samples for immunohistochemistry analysis were taken. (**B**) Timeline of the experimental setup. Animals were submitted to t-BCCAO and treated with ELF-EMS (13.5 mT/60 Hz) 1 h after injury and the following three consecutive days. Behavioral tests were conducted at 1, 3 and 7 days after surgery and brain samples were taken for immunostaining. On day 7, locomotion and anxiety were assessed using the open field and elevated plus-maze test respectively. (**C**) Neurological score was determined 1, 3 and 7 days after injury. A higher score represents a higher number of deficits. (**D**) Rearing frequency in the open field during 10 min. (**E**) Quantification of time spent in the open areas (open arms and center) seven days after surgery. n = 8–10 per group. Data is shown as mean  ±  SEM, * *p*  <  0.05, ** *p*  <  0.01 as determined using *t*-test for unpaired samples. Dotted lines showing non-ischemic non-treated animal behavior.

**Figure 2 ijms-24-11117-f002:**
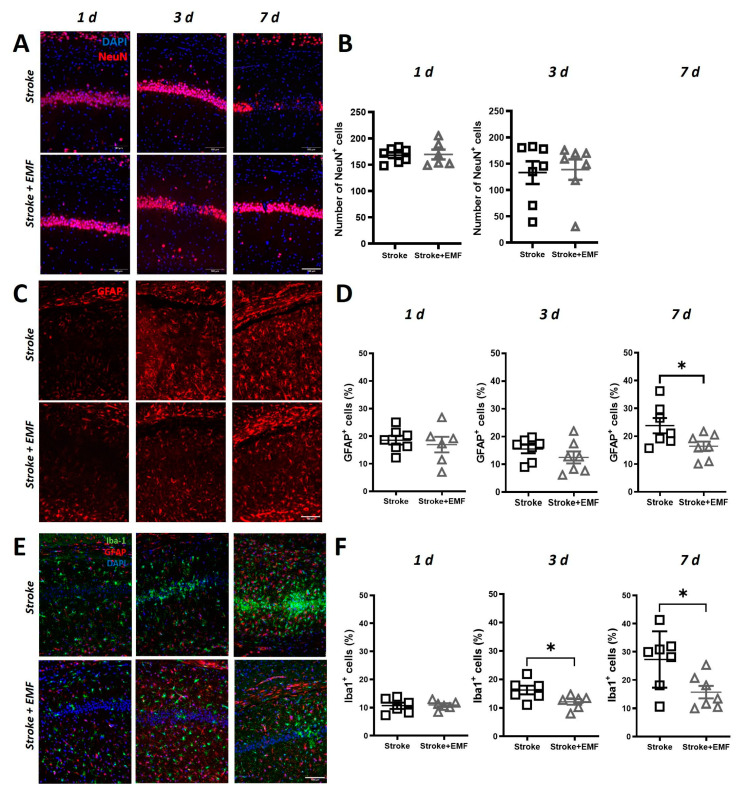
ELF-EMS affects neuronal survival and glial activation in the hippocampus of gerbils submitted to transient bilateral common carotid artery occlusion (t-BCCAO) (**A**) Images show immunofluorescence staining of brain samples with neuronal marker (NeuN) in red and nuclear counterstain (DAPI) in blue at 1, 3 and 7 days (n = 6–7/group). (**B**) Quantification of the number on NeuN+ in the CA1 and CA2 region of the hippocampus. (**C**) Immunofluorescence staining of brain samples with astrocytes (GFAP) marker in red at 1, 3 and 7 days (n = 5–8). (**D**) Quantification of the GFAP+ cells percent area in the CA1 and CA2 region of the hippocampus. (**E**) Immunofluorescence staining of brain samples with microglia/macrophage (Iba-1) in green, astrocytes (GFAP) markers in red and nuclear counterstain (DAPI) in blue at 24 h, 3 and 7 days (n = 5–7). (**F**) Quantification of the Iba1+ cells percent area in the CA1 and CA2 region of the hippocampus. Scale bar = 100 µm. Statistical analysis was made with *t*-student test for unpaired samples * *p* < 0.05.

**Figure 3 ijms-24-11117-f003:**
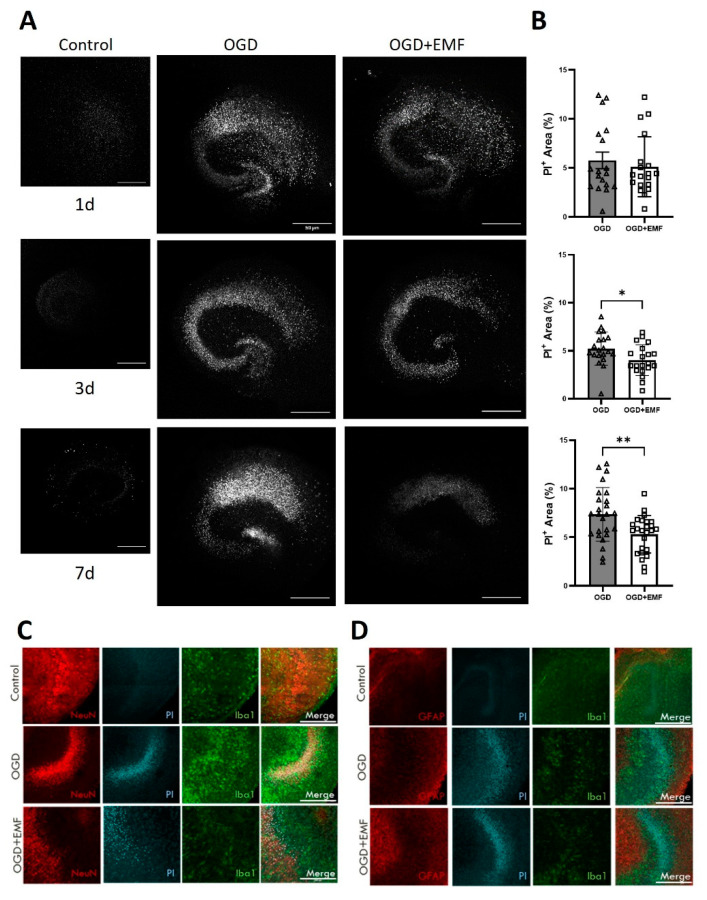
ELF-EMF improves neuronal survival in mouse organotypic hippocampal slices culture after OGD/R injury (**A**) Images of mouse OHSC with PI+ cells after 1, 3 and 7 days of reperfusion. Control pictures show viability of the slices. Scale bar = 50 µm. (**B**) Quantification of the number on PI+ in the OHSC (n = 15–25). (**C**) Immunofluorescence staining of mouse OHSC with neuronal (NeuN) in red, PI in cyan and microglia (Iba1) markers in green at day 7 after I/R. (**D**) Immunofluorescence staining of mouse OHSC with astrocytes (GFAP) in red, PI in cyan and microglia (Iba1) markers in green at day 7 after I/R to show that PI+ cells are indeed neurons. Scale bar = 200 µm. Statistical analysis was made with *t*-student test for unpaired samples; * *p* < 0.05, ** *p* < 0.01.

**Figure 4 ijms-24-11117-f004:**
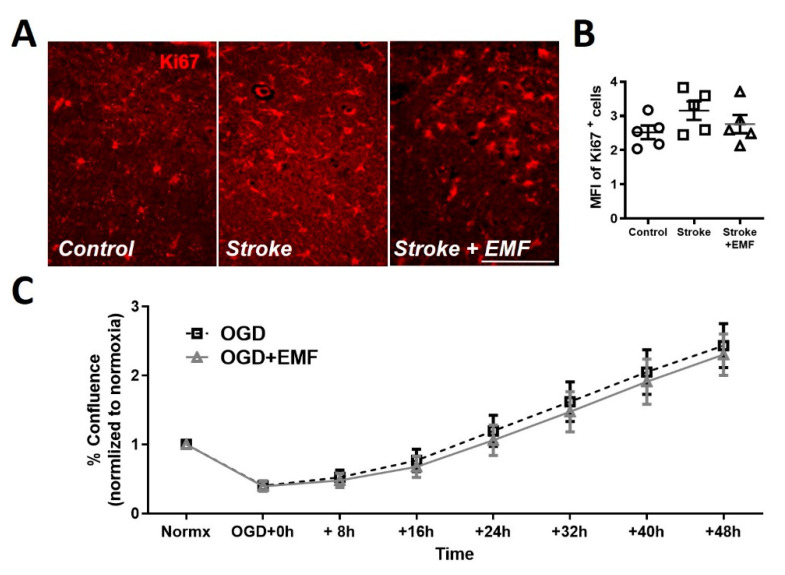
ELF-EMS does not affect microglial proliferation in an ischemia/reperfusion model. (**A**) Immunofluorescent staining of brain samples with proliferation marker (Ki67) in red at day 1 in the hippocampus stratum radiale. Scale bar = 100 µm. (**B**) Quantification of median fluorescent intensity (MFI) of Ki67+ cells (n = 5). (**C**) Quantification of BV2 cells confluence (%) during 48 h of reperfusion after 3 h of OGD. The data represent the mean ± SE. Statistical analysis was made using One-way ANOVA with Bonferroni post-hoc.

**Figure 5 ijms-24-11117-f005:**
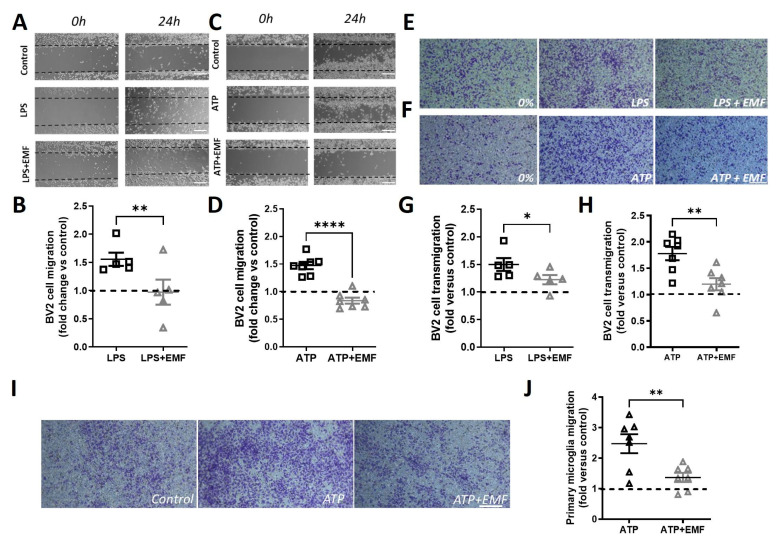
ELF-EMS decreases ATP-and LPS-induced migration and transmigration of microglia. In BV2 cells, scratch assay was made in serum-free cells cultures using Ibidi silicone inserts and migration was quantify 24 h after. For transmigration, transwell inserts (8 µm pore size) were used and LPS (100 ng/mL) or ATP (300 µM) was added to the bottom chamber. Transmigration was quantified 6 h after stimulation. (**A**,**C**). Images of the scratch at 0 h (i.e., immediately after scratching) and 24 h after LPS (100 ng/mL, (**A**)) or ATP (300 µM, (**C**)). (**B**,**D**) Quantification of LPS-induced BV2 cells migration to the scratch area reported as fold-change relative to control. (**E**) Images of the migrating BV2 cells (bottom chamber) 6 h after adding LPS (100 ng/mL) with serum-free medium used as control. (**F**) Images of the migrating BV2 cells (bottom chamber) 6 h after adding ATP (300 µM) with serum-free medium used as control. (**G**) Quantification of LPS-induced BV2 cells transmigration to the bottom chamber reported as fold versus control. (**H**) Quantification of ATP-induced BV2 cells transmigration to the bottom chamber reported as fold versus control. (**I**) ELF-EMS decreases ATP-induced transmigration on mouse primary microglia cells. Representative images of the migrating microglia cells (bottom chamber) 24 h after adding ATP (300 µM), culture medium was used as control. ATP + EMF group was treated with ELF-EMS (13.5 mT/60 Hz, 20 min) after seeding. (**J**) Quantification of ATP-induced mouse primary microglia transmigration to the bottom chamber reported as fold versus control. Scale bar = 100 µm. The data represent the mean ± SE from n = 5–7 independent experiments; * *p* < 0.05, ** *p* < 0.01, **** *p* < 0.0001, *t* student for unpaired samples.

**Figure 6 ijms-24-11117-f006:**
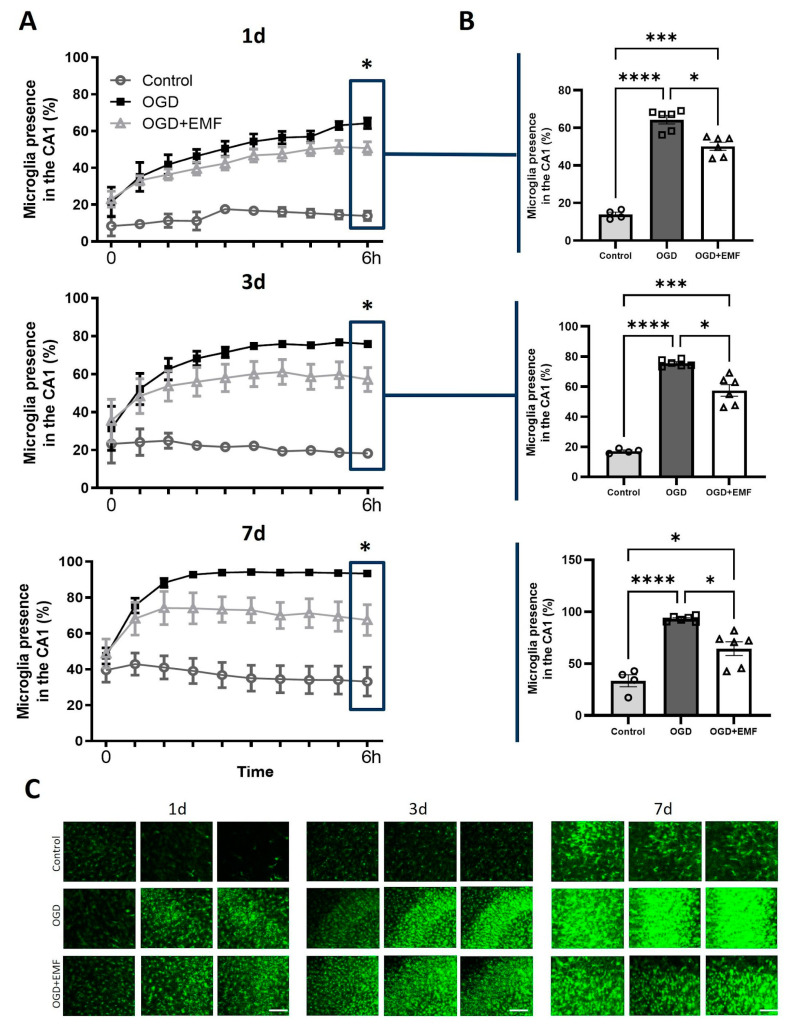
ELF-EMS decreases microglia migration in organotypic hippocampal slice cultures (OHSC). OHSC from CX3CR1WT/eGFP mice were submitted to OGD and stimulated with ELF-EMS for four consecutive days. Ten images were taken during a 6 h period at day 1, 3 and 7 after OGD for quantifying microglia migration. (**A**) Quantification of microglial density in the CA1 area of the hippocampus in 10 time points during 6 h. Statistical analysis was done with Two-way ANOVA, Tukey (post-hoc) * *p* < 0.05, n = 4–6. (**B**) Quantification of microglia presence in the CA1 region of the hippocampus in the last time point during microglia migration monitoring. Statistical analysis was done with One-way Anova, Tukey (post-hoc) * *p* < 0.05, *** *p* < 0.01, **** *p* < 0.001, n = 4–6. (**C**) Representative images of microglia in green in the CA1 area of the hippocampus in the last time point during microglia migration monitoring at 1, 3 and 7 days. Scale bar = 10 µm.

**Figure 7 ijms-24-11117-f007:**
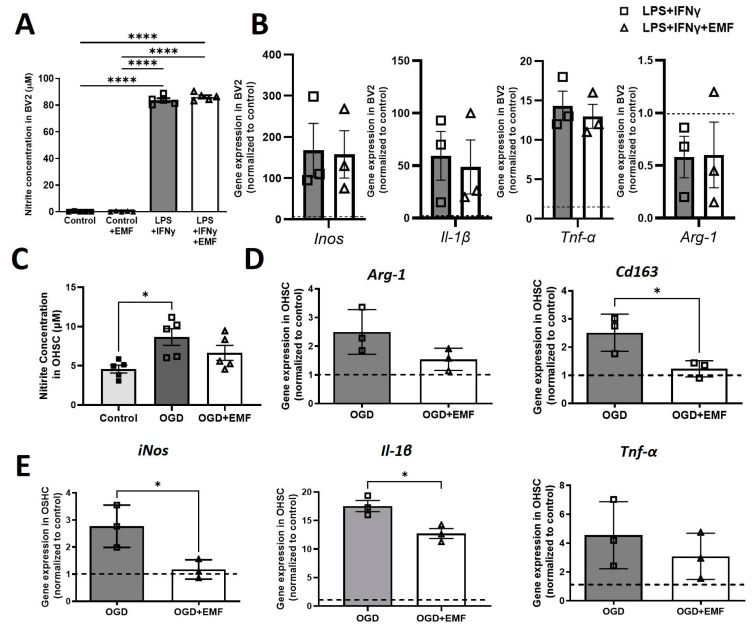
ELF-EMS treatment affects inflammatory markers in OHSC but not in BV2 cells. BV2 cells were stimulated with LPS and IFNγ. OHSC were submitted to 1 h OGD and 7 days of reperfusion. (**A**) Quantification of nitrite concentration as an indirect measure for NO (n = 3) in BV2 cell cultures. (**B**) mRNA expression of pro-inflammatory signature genes (*Inos*, *Il-1β* and *Tnf-α*) and anti-inflammatory signature gen *Arg-1* in BV2 cell cultures (n = 3). (**C**) Quantification of nitrite concentration as an indirect measure for NO in mouse OHSC (n = 5) (**D**) mRNA expression of anti-inflammatory signature genes *Arg-1* and *Cd163* in mouse OHSC. (**E**) mRNA expression of pro-inflammatory signature genes (*iNos*, *Il-1β* and *Tnf-α*. n = 3 (7 slices/n) in mouse OHSC. Statistical analysis was done with One-way Anova, Tukey (post-hoc) or *t*-test. * *p* < 0.05, **** *p* < 0.0001.

**Table 1 ijms-24-11117-t001:** Primers for qRT-PCR.

Genes		Primers (5′-3′)
*Gapdh*	Forward	ACCACAGTCCATGCCATCAC
	Reversed	TCCACCACCCTGTTGCTGTA
*Actin b*	Forward	GGCTGTATTCCCCTCCATCG
	Reversed	CAGTTGGTAACAATGCCATGT
*iNos*	Forward	CCCTTCAATGGTTGGTACATGG
	Reversed	ACATTGATCTCCGTGACAGCC
*Il-1β*	Forward	ACCCTGCAGCTGGAGAGTGT
	Reversed	TTGACTTCTATCTTGTTGAAGACAAACC
*Tnfα*	Forward	CCAGACCCTCACACTCAG
	Reversed	CACTTGGTGGTTTGCTACGAC
*Arg-1*	Forward	GTGAAGAACCCACGGTCTGT
	Reversed	GCCAGAGATGCTTCCAACTG
*Cd163*	Forward	GCTAGACGAAGTCATCTGCACTGGG
	Reversed	TCAGCCTCAGAGACATGAACTCGG

## Data Availability

The data presented in this study are available on request from the corresponding authors.

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
