# Peer review of "Extremely Low-Frequency Electromagnetic Stimulation (ELF-EMS) Improves Neurological Outcome and Reduces Microglial Reactivity in a Rodent Model of Global Transient Stroke"

_ijms, 2023, doi:10.3390/ijms241311117_

Round 1

Reviewer 1 Report

Dear Authors,

I have read with great interest and pleasure your manuscript and acknowledge the therapeutic potential of this non-invasive procedure. The manuscript is, in my opinion, well written and clear, addressing all the potential mechanisms of action on neurons, astrocytes and microglia. As pointed out in the manuscript, technical details on magnetic field induction and frequency will need to be worked out by further research.

I was puzzled, though, by your statement in line 294, that EMF increased microglial activation. Would we not want to decrease activated microglia in the acute phase post-stroke? Further, given the dual nature of neuroinflammation, with deleterious effects in the acute phase, but whith microglia then switching and polarizing toward a neuroprotective phenotype (M2) and promoting the tissue repair processes in the subacute and chronic phases, I wonder whether ELF-EMF could allow for only a moderate inhibition of microglia activation and migration in the acute phase. I mean - systemic lymphocyte depletion in acute stroke proved to worsen the neurological outcome, indicating that a certain degree of inflammation is a normal response to the tissue injury and should be allowed. 

As for the language, I suggest minor spelling and Typo corrections:

- line 31: dead - replaced with death

- line 32-33 - rephrased

- line 33 - treatment strategies (without the "S" in treatments

- line 36 - cloth - replaced with clot

- line 63 - underlaying - replaced with underlying

- in line 241 - did you mean iNOS instead of Inos?

- line 268 - is the word "ischemia" missing in transient global cerebral (ischemia)?

lines 350-351 - rephrased

line 358 - gaiting - replaced with "gating"

Author Response

Comment of the reviewer: "I was puzzled, though, by your statement in line 294, that EMF increased microglial activation. Would we not want to decrease activated microglia in the acute phase post-stroke? Further, given the dual nature of neuroinflammation, with deleterious effects in the acute phase, but whith microglia then switching and polarizing toward a neuroprotective phenotype (M2) and promoting the tissue repair processes in the subacute and chronic phases, I wonder whether ELF-EMF could allow for only a moderate inhibition of microglia activation and migration in the acute phase. I mean - systemic lymphocyte depletion in acute stroke proved to worsen the neurological outcome, indicating that a certain degree of inflammation is a normal response to the tissue injury and should be allowed."

Our answer: We like to thank the author for this remark and insight. In the original manuscript, we have following statement in line 294: “Similar to astrocytes, continued exposure to EMF for over 7 days increased microglia activation in models of transient global stroke [18]. We confirmed the effect of ELF-EMS on microglia reactivity in OHSC.” By this, we mean that in contrast to our results, the study developed by Raus et al. [18], also found an increase in microglia reactivity after seven days of exposure to EMF, pointing out the importance of the exposure time in the effects achieved by EMF. We thank the reviewer for highlighting this sentence, and to better present our analysis, we rewrite it as follows: “However, similar to astrocytes, Raus et al. found that continued exposure to EMF for over seven days increased microglia activation in models of transient global ischemic stroke [18]. This highlights the differential effect of EMF depending on its settings (such as fre-quency, duration, time of application after stroke, magnetic flux density,..). Moreover, we confirmed the decreasing effect of 4 sesions of 20 min of ELF-EMS on microglia reactivity in OHSC”

As we presented, ELF-EMS has an effect decreasing some inflammatory M1 markers such as IL1beta and iNOS (in the OHGS see figure 7E). It also lowered the expression of M2 markers. Therefore, we hypothesize that our EMF is a modulator of the inflammatory response and microglia reactivity. As depicted in figure 7, EMF reduces the levels of inflammatory markers, but never to the levels of control conditions, meaning there is a moderate effect. We do agree with the reviewer that more research, e.g. adjusting the time of application of EMF or the modes of EMF, are required to further optimize this treatment.

As nicely stated by the reviewer, the role of microglia is a double-edged sword, meaning they have inflammatory but also regenerative properties. In the applied t-BCCAO stroke model, reactive microglia in the hippocampus have been proven to worsen stroke outcome by other authors. Moreover, as we have explained in the discussion section, it has been found that glia depletion protected against neuronal damage in an infrasound-induced neuronal damage model, and specifically microglia depletion resulted in a significant decrease in ischemic infarct volume, relieves inflammation and improves myelination of white matter in the brain, which prevents cognitive decline in diabetic animals.

Given these contradictory results, a lot of further research is needed. Single cell transcriptomics /proteomics research is needed to have insight in the phenotypes of microglia and also lymphocytes after stroke. We also need to map precisely when in the regenerative process after stroke these microglia are activated and what their exact contribution to the healing process is. In addition, the precise type of ischemic stroke (dMCAO, tMCAO, t-BBCAO,) should be taken in consideration.

We thank this reviewer very much for his thorough correction of spelling and grammar and we corrected all these errors.

Reviewer 2 Report

This is in the main a well presented paper, that details some very interesting  data on the effect of "Extremely Low-Frequency Electromagnetic Stimulation (ELF- 2 EMS) Improves Neurological Outcome and Reduces Microglial 3 Reactivity in A Rodent Model of Global Transient Stroke" i.e. precisely what the title states. The study presents a diversity of studies to investigate and answer this question, in response to which I have only some minor comments to make. The precise mechanism(s) ELF-2EMS mediates its effects are not presented and the authors do not claim that they have delineated these. The diversity of the results presented are however would be interesting to many readers.

My comments.

1. Avoid the use of the claim that your results "prove"-this should be "significantly demonstrate" or similar 

2. Figure 1 presents results as mean +/- SEM and comparisons are by Mann-Whitney test. Given that the Mann-Whitney compares medians and not means- this is not acceptable. The authors should present this (and any other similar) data as medians and IQ ranges, or alternativley use a two-sample t-test for comparisons.

3. Some minor English editing required.

As above

Author Response

  1. Avoid the use of the claim that your results "prove"-this should be "significantly demonstrate" or similar 

Thank you for your suggestion; We adapted this in the abstract on line 13, in the discussion on line 301 and line 332

  1. Figure 1 presents results as mean +/- SEM and comparisons are by Mann-Whitney test. Given that the Mann-Whitney compares medians and not means- this is not acceptable. The authors should present this (and any other similar) data as medians and IQ ranges, or alternativley use a two-sample t-test for comparisons.

Thank you very much for this clarification. We have redone the statistical analysis thereby applying a two-sample t-test of figure 1. The text and figure legend was adapted accordingly.

  1. Some minor English editing required.

We have thoroughly reviewed the text and demarked the changes in yellow.

Reviewer 3 Report

The authors have demonstrated that extremely low-frequency electromagnetic stimulation improves neurological outcome and reduces microglial reactivity in a rodent model of global transient stroke. The contents are interesting; however, some data have problems that should be improved.

1.     In figure 1C, D, F, and figure 6B, the data points should be shown.

2.     It is not described how the sample size was determined. For example, the n number of figure 3B is 15-20; but it of figure 7 is only 3. The explanation is needed. Also, the n number must be at least 5.

3.     The control in figure 7D and E does not have dispersion.

4.     Some images do not have scale bar.

5.     The animal number used in all experiments is not described.

Round 2

Reviewer 3 Report

Although I remain concerned about the low n number, good reason was explained, so I recommend the acceptance.